



# Development and calibration of a global hydrological model for integrated assessment modeling

Tingju Zhu[1], Petra Döll[2,3], Hannes Müller Schmied[2, 3], Claudia Ringler[1], Mark W. Rosegrant[1]

[1]International Food Policy Research Institute, 1201 Eye St., NW, Washington, DC, 20005-3915, USA

[2]Institute of Physical Geography, Goethe-University Frankfurt, Altenhöferallee 1, 60438 Frankfurt, Germany

[3]Senckenberg Biodiversity and Climate Research Centre (BiK-F), Senckenberganlage 25, 60325 Frankfurt, Germany

*Correspondence to*: Tingju Zhu (t.zhu@cgiar.org)

**Abstract.** This paper describes the IMPACT Global Hydrological Model (IGHM), a component of the International

Model for Policy Analysis of Agricultural Commodities and Trade (IMPACT) integrated modeling system. IMPACT has been developed in the early 1990s to identify and analyze long-term challenges and opportunities for food, agriculture, and natural resources at global and regional scales and builds on a series of previous food demand and supply projections models developed at the International Food Policy Research Institute since the early 1980s. The IGHM has been developed to assess water availability and variability as drivers of water use and irrigated crop

production in IMPACT. It adopts a saturation runoff generation scheme and uses a linear groundwater reservoir to simulate base flow in 0.5º latitude by 0.5º longitude grid cells over the global land surface excluding Antarctica. The IGHM has four cell-specific calibration parameters, which are determined through maximizing the Kling–Gupta efficiency (KGE) with a genetic algorithm at the grid cell level, using gridded natural runoff series generated by the WaterGAP Global Hydrological Model (WGHM). During the calibration and validation periods, globally, the

majority of grid cells attain KGE values greater than 0.50.  As a meta-model of the more computationally expensive WGHM, IGHM transfers the climate-hydrology dynamics provided by WGHM into the integrated IMPACT model at a lower computational cost and enables coupling hydrology and other related processes considered in IMPACT which are important for analyzing long-term water and food security under a range of environmental and socioeconomic changes.

**1 Introduction**

Short-term variation and long-term trends in water availability strongly affect water supply and food production (Rosegrant et al. 2009). Analyzing such effects at a global scale requires integrated assessment modeling that brings together global hydrological, agronomic, and economic modeling capabilities into a coherent framework. To evaluate the food security effects of variations in water availability, alternative irrigation technologies, and water

management strategies, in the late 1990s a river-basin level, global water resource model was developed and linked to the International Model for Policy Analysis of Agricultural Commodities and Trade (IMPACT), leading to the integrated water and food projections model IMPACT-WATER (Rosegrant et al. 2002). The original IMPACT-WATER model optimizes water supplies at the river basin or sub-river basin level, considering reservoir regulation as well as constraints of water balance, infrastructure capacity and environmental flow requirements (Cai and




Rosegrant 2002). It used historical runoff simulated by an early version of the WaterGAP model (Döll et al., 2001; Döll et al., 1999). More recently, to assess coupled hydrological, water resource, agronomic, and agroeconomic responses to climate change, we developed a distributed, conceptual global hydrological model to integrate water availability simulation capability into an updated IMPACT modeling system (Robinson et al. 2015; Zhu et al. 2013;

Zhu and Ringler 2012); this new module is called the IMPACT Global Hydrological Model (IGHM) and is the focus of this article. The IGHM simulates natural runoff without considering human activity impacts on terrestrial water resources. This model runs at a monthly time step and has a spatial resolution of 0.5º C latitude by 0.5º C longitude, which is about 3,100 km$^2$ near the equator and covers the entire global land surface, excluding Antarctica. Previous versions of IGHM have been used in a range of policy analyses, such as assessing adaptation options to climate

change and the contribution of irrigation to regional and global food securtiy (Calzadilla et al. 2013; Liu et al. 2014; Nelson et al. 2010; You et al. 2011) and have also contributed to the development of other macroscale water resources models, such as Strzepek et al. (2013). The following sections review the main characteristics of existing global hydrological models, followed by a brief discussion on the challenges faced by macroscale hydrological modeling and the choice of a parsimonious structure for the IGHM model, which operates as a module in the larger

IMPACT modeling system.

## 1.1 Review of global hydrological models

Several global hydrological models have been built since the 1980s. The characteristics and performance of these models are discussed in several review papers (Bierkens, 2015; Sood and Smakhtin, 2014) and multimodel assessments of global hydrological regimes, water resources, and water scarcity (Haddeland et al., 2011; Schewe et

al., 2014; Veldkamp et al., 2017). Here, we compare the main characteristics of six existing global hydrological models that are more relevant to IGHM because of their uses of conceptual rainfall–runoff modeling procedures (Table 1). These six models have varying degrees of complexity in model structure and the number of parameters. There are also other global hydrological models that are not included in this review because they are either land process models, or fully couple water and energy balances.

Among these six global hydrological models, four use a daily time step. The exceptions are the WBM model (Vörösmarty et al. 1989, 1998), which uses a semi-monthly time step, and the WASMOD-M model (Widén-Nilsson et al. 2007, 2009), which uses a monthly time step. Models that run at a daily time step previously used various techniques to downscale monthly average meteorological input data to a daily scale, due to a lack of consistent daily global climate databases at that time; some models generated synthetic daily precipitation using the number of wet

days in a month and a statistical model that produces the distribution of wet days (Arnell 1999; Döll et al. 2003; Gerten et al. 2004; Wisser et al. 2010), whereas others used daily distributions from climate model reanalysis to disaggregate monthly average precipitation to daily values (Van Beek and Bierkens 2008).





**Table 1**. Main characteristics of selected existing global hydrological models.

| Model | Time step | Climate input (in brackets, ISIMIP climate forcing ) | PET scheme | Runoff scheme | Number of tunable parameters | Data used in calibration (in brackets, number of gauging stations) | Parameter determination | References |
|---|---|---|---|---|---|---|---|---|
| LPJmL | Daily | P, T, C, and WD (P, T, SD, LD) | Priestley–Taylor | Saturation excess | n.a. | n.a. | Biophysical spatial database | (Gerten et al., 2004) |
| Mac-PDM | Daily | P, T, V, C, W SH, and WD | Penman/ Penman–Monteith/ Priestley–Taylor | Saturation excess | n.a. | n.a. | Biophysical spatial database | (Arnell 1999, 2003; Gosling and Arnell, 2011) |
| PCR-GLOBWB | Daily | P, T, V, C, W | Penman–Monteith | Saturation excess | n.a. | n.a. | Biophysical spatial database | (Van Beek and Bierkens, 2008; Van Beek et al., 2011) |
| WASMOD-M | Monthly | P, T, and V | Air temperature and relative humidity-driven function | Linear reservoirs for slow and fast runoff | 5 | Long-term average discharge (654 gauging stations) | Identify "acceptable runoff" from simulations of globally fixed parameter-value combination sets (constant parameter values within a basin); and apply regionalized parameter sets to ungauged basins | (Widén-Nilsson et al., 2007, 2009) |
| WBM | Semi-monthly/ daily | P, T, V, SD, W, and daily temperature range | Priestley–Taylor | Saturation excess | n.a. | n.a. | Biophysical spatial database | (Fekete et al., 2002; Vörösmarty et al., 1989, 1998; Wisser et al., 2010) |
| WaterGAP | Daily | P, T, C, SH, and WD (P, T, SD, LD) | Priestley–Taylor | Non-linear function of soil moisture | 3 | Long-term average discharge over 4 to 30 measurement years (1319 gauging stations) | Biophysical spatial database; runoff coefficient for tuning basins, and applying regionalized parameters to ungauged basins | (Döll and Fiedler, 2008; Döll et al., 2003, 2009; Hunger and Döll, 2008; Müller Schmied et al., 2014, 2016) |

Notes: P = precipitation; T = temperature; C = cloudiness; SH = average daily sunshine hours; WD = number of wet days in a month; W = wind speed; SD = shortwave downward radiation; LD = longwave downward radiation. The ISIMIP climate forcing data can be found at www.isimip.org





Before the WATCH Forcing Data (WFD) (Weedon, et al. 2011) was developed under the European FP6- funded Water and Global Change (WATCH) project (http: //www.eu-watch.org), the majority of global hydrological models used gridded monthly climate forcing of the CRU TS 2.1 database developed by the Climatic Research Unit (CRU) at the University of East Anglia (Mitchell and Jones 2005) or other versions of the CRU database (Arnell,

1999; Van Beek and Bierkens, 2008; Döll et al., 2003; Gerten et al., 2004; Widén-Nilsson et al., 2007; Wisser et al., 2010). The primary consideration is that the CRU data are based on observations covering the global landmass and were processed in a consistent manner (Van Beek and Bierkens 2008). Typical meteorological variables used by those models include precipitation, temperature, water vapor pressure, cloudiness, hours of sunshine, and number of wet days in a month. Among the six models, only LPJmL simulates vegetation dynamics, and couples it with

hydrological process simulation. Nowadays most models use readily available daily climate forcing such as the WFD (Weedon, et al. 2011) or the more recent WATCH Forcing Data methodology applied to ERA-Interim (WFDEI) data (Weedon et al. 2014).

All six models in Table 1 use a degree-day approach to simulate snow accumulation and melt, based on air temperature. The specific treatment varies across models. For instance, the WBM uses an empirical temperature-

and precipitation-based formula; the WASMOD-M uses an exponential function of air temperature to determine snow accumulation and melt; whereas the PCR-GLOBWB adopts the snow scheme in the HBV model, which specifies that snow melts before glaciers, and different degree-day factor values are applied to snowpack and glaciers.

The runoff schemes of four global hydrological models in Table 1 (Arnell 1999; Van Beek and Bierkens 2008;

Gerten et al. 2004; Vörösmarty et al. 1989), are based on the saturation excess concept, although their specific mathematical formulations of runoff generation vary. In contrast, the WaterGAP Global Hydrological Model (WGHM) uses a "beta function", which is nonlinear in soil moisture, to produce total runoff from land that is subsequently partitioned into fast surface runoff, subsurface runoff, and groundwater recharge (Döll et al. 2003). Finally, the WASMOD-M model specifies slow and fast runoff as flows from two connected linear reservoirs

(Widén-Nilsson et al. 2007). Among the four models that adopt a saturation excess runoff scheme, both the Mac-PDM model (Arnell 1999; Gosling and Arnell 2011) and the PCR-GLOBWB model (Van Beek and Bierkens 2008) assume that runoff is generated by liquid precipitation falling on fully saturated soil, and the soil moisture storage capacity varies statistically within a grid cell. The PCR-GLOBWB model adopts the linear reservoir concept to model discharge from groundwater storage, with outflow being a linear function of active groundwater storage.

Parameters in all these models are related to biophysical properties. Only in the WGHM and WASMOD-M models some parameters are adjusted in a basin-specific manner by calibrating against observed river discharge. Comparisons of simulated runoff with measured discharge are primarily for validation purposes for the other models (Van Beek and Bierkens 2008; Gerten et al. 2004; Gosling and Arnell 2011; Vörösmarty et al. 1998). More recently, the roles of dam operation and river routing schemes in global hydrological models were investigated by comparing

simulated river discharge against observed values at gauging stations using multimodel comparisons (Masaki et al.





2017; Zhao et al. 2017). It is worth noting that the shape parameter *b* in the PCR-GLOBWB model is estimated based on the distribution of maximum rooting depths within each cell (Van Beek and Bierkens 2008).

In the parameterization of the WASMOD-M model, Widén-Nilsson et al. (2007) conduct simulations using 1,680 globally fixed parameter-value sets (constant parameter values within a basin), generated through combining the
discrete values of five model parameters, and determine the best-performing set by identifying "acceptable runoff" from those simulations. In the WGHM model, a single parameter called runoff coefficient in the vertical water balance is tuned (and possibly 1-2 correction factors) by comparison to mean annual river discharge at 1319 stations worldwide, in order to avoid overparameterization and to make tuning feasible in a large number of river basins (Döll et al. 2003; Hunger and Döll 2008). Within each basin, the runoff coefficient is homogeneously adjusted for
all cells to achieve reasonable fit to observed river discharge. Compared to previous WGHM model versions, WaterGAP 2.2b allows an uncertainty of 10% of long-term average observed river discharge (following Coxon et al, 2015) so that calibration runs in four steps: 1) test if runoff coefficient alone is enough to calibrate the uncertainty to below 1% of observed value; 2) test if runoff coefficient alone is enough to calibrate when 10% uncertainty of observed values are allowed; 3) adapt observed value by 10%, and test if runoff coefficient plus runoff correction
factor (applied universally to all cells in the basin, to ensure that the simulated long-term average discharge is sufficiently close to observation,) are sufficient for calibration; and 4) add station correction factor (multiplier to fit simulated to the observed value in order to avoid error propagation in the downstream basin) (Müller Schmied et al., 2014, Müller Schmied 2017). Elsewhere, to derive global long-term average runoff fields using the WBM model, Fekete et al. (2002) also apply a runoff correction factor which equals the ratio of measured and simulated long-term
average discharge, without tuning model parameters. For both WGHM and WASMOD-M, the long-term average of measured discharge is used in parameter tuning in order to minimize the effects of human regulation, which are more pronounced in an intra-annual time scale (Döll et al. 2003; Hunger and Döll 2008).

Large-scale hydrological modeling is influenced by several factors. Among them, input data including climate forcing and parameters appear to be of the highest importance, whereas model structure seems to be least important,
and the influence of spatial and temporal discretization lies in between (Arnell 1999; Döll et al. 2008). Haddeland et al. (2011) compared global scale simulation results of five global hydrological models and six land surface models driven by a single global meteorological dataset for the period 1985–1999. The WGHM model-simulated runoff appears to be closer to the observations than all other model results, because it is the only model among all participating models that is calibrated. Döll et al. (2003) state that imperfect runoff simulations can be caused by
several factors, ranging from input data to modeled processes, and parameter tuning against measured recharge, which provides additional information, can improve model performance. Müller Schmied et al. (2016a, 2016b) applied five state-of-the-art climate input data to WGHM and showed reasonable differences in model outputs, especially at the grid cell level, as well as in non-calibrated regions, which shows that the choice of climate input data is very important for water resources assessment.



### 1.2 Challenges of macroscale hydrological modeling and considerations for linking to an integrated assessment model

A macroscale hydrological model is a generalization or extrapolation of a catchment-scale model (Arnell 1999). Nevertheless, macroscale hydrological modeling faces several challenges which are unique when compared against those faced by watershed scale hydrological models. Döll et al. (2016) identified seven challenges in the development and application of global hydrological models, ranging from data scarcity for quantifying human water use and uncertainties in climate forcing input data, to uncertainties in vegetation responses to changing climate and atmospheric $CO_2$ concentrations, to discrepancies in simulated hydrological responses to climate change, and to groundwater simulation in and water scarcity and human interference assessment with global hydrological models.

As a parsimonious global scale model that is focused on simulation of natural runoff, the IGHM faces primarily two challenges: data constraints and, compared with watershed scale hydrology, our more limited understanding of macroscale hydrological processes. These two challenges are interlinked. First, global hydrological models are constrained by the availability and quality of global data sets (Widén-Nilsson et al., 2009; Müller Schmied et al., 2016). New data sets and updated versions of existing data sets of climate, soil, and water bodies are being made available frequently, enabling improved quantification of hydrological variables; however, the representativeness and quality of these data sets are fundamentally limited by available in situ observations (Harris et al. 2014; Lehner et al. 2011). Second, hydrological models are traditionally developed based on measurements and understanding of "micro" scale processes. As such, observed data and hydrological processes are often not compatible or representative at larger scales relevant for macroscale processes (Singh and Woolhiser 2002). Therefore, sophisticated data-intensive watershed hydrological models may not be suitable for macroscale hydrological modeling, due to their large data requirements (Chen et al. 2007), the relatively highly detailed specifications of hydrological processes with a sophisticated model structure, and the large number of parameters that are tailored for a specific watershed at the cost of broader model applicability.

Moreover, a macroscale hydrological model like IGHM, which operates as a module in a larger integrated assessment model, should avoid needless process complexity, while representing all key hydrological processes. Hydrological processes and model parameters are scale-dependent; however, as scale increases, spatial aggregation of data leads to loss of information (Turner 1990). A less detailed conceptual model is found to simulate long-term water balance more reliably as a watershed goes up in scale (Merz et al. 2009). For the long-term assessment of the impacts of climate change on global water and food trade, a monthly temporal scale is sufficient and may simplify the calibration process (Sood and Smakhtin 2014). In order to develop a generic form that is applicable to a broad spectrum of systems across the globe and capable of characterizing the diverse nature of climate and hydrology over space and time, macroscale hydrologic models must be simplified (Dooge 1986; Vörösmarty et al. 1989; Sivakumar 2004). Macroscale hydrologic models should be simple in structure and parsimonious in parameters, focusing on dominant processes that can be constrained by available data. This echoes the finding that parsimonious models are preferable for data-scarce regions (Pande et al. 2011). Moreover, a notable advantage of parsimonious models is the convenience of linking to or coupling with integrated assessment models.



Using priori parameters in macroscale hydrological models without calibration result in sub-optimal runoff simulation (Beck 2016). To increase the realism of water resources simulated by the parsimonious IGHM, with a monthly time step and without lateral routing, its parameters were adjusted, individually for each grid cell, against

grid cell runoff computed by the more complex and calibrated WGHM that uses a daily time step. It is preferable to calibrate IGHM in this way for each of approximately 67000 grid cells than to calibrate it against observed river discharge available for a necessarily much smaller number of discharge gauging stations as 1) river routing and the impact of water use on river discharge is not simulated by IGHM and 2) the more complex WGHM can be assumed to be better able to distribute observed mean annual discharge to upstream grid cells than the simpler IGHM.

Calibrating a parsimonious model to simulated output of more sophisticated models has been done elsewhere previously. The simple carbon cycle–climate model MAGICC6, which is the core model of the integrated assessment model IMAGE 3.0 (Stehfest et al., 2014) was calibrated against the higher complexity atmosphere–ocean general circulation models (AOGCMs) and carbon cycle models (Meinshausen et al., 2011b). After being

calibrated, MAGICC6 can emulate, with considerable accuracy, globally aggregated characteristics of these more complex models (Meinshausen et al., 2011a).

## 2 IMPACT Global Hydrological Model

### 2.1 Data

Climate data that drive the IGHM model include precipitation, average temperature, and downward shortwave

radiation, by month. In this article, these data are aggregated from daily to monthly fields for the period 1980–2009, using the reanalysis-based Watch Forcing Data based on ERA-Interim (Weedon, et al. 2014).

The calculation of potential evapotranspiration (PET) also requires elevation, for which the 0.5º resolution global gridded elevation database from the CRU is used. The gridded albedo values are from the WGHM model database, as used in Müller Schmied et al. (2014), which is based on the 2004 MODIS land cover classes (IGBP

classification). Open water bodies considered in the IGHM model include lakes and wetlands, as used in Müller Schmied et al. (2014, 2016), which is based on the Global Lake and Wetland Database (GLWD) (Lehner and Döll 2004) and the Global Reservoir and Dam Database (GRanD) (Lehner et al. 2011) for the areas of natural lakes and wetlands, and the areas of manmade reservoirs, respectively.

WGHM has been parameterized using observed river discharge around the world, preferably during the period

1971–2000. The calibration of the WGHM model against observed long-term average river discharge significantly reduces the impact of climate, forcing uncertainty on estimated runoff (Müller Schmied et al. 2014, 2016). In this paper, we use gridded monthly natural runoff generated in 0.5° grid cells during 1980–2009 simulated by the WGHM in its version 2.2b (Müller Schmied et al. 2016) to determine parameters of the IGHM model for each 0.5°





grid cell separately. The "natural runoff" refers to monthly runoff simulated by the WGHM in a setup which does not consider human water uses and reservoir operation.

### 2.2 Model structure

The IGHM model is a parsimonious conceptual global hydrological model. It simulates the water balance for each
grid cell independently, treating each grid cell as an individual catchment. As illustrated in Fig. 1, the model uses a temperature-index method to simulate snow accumulation and melting, and determines total saturation excess runoff with a probabilistic distribution of soil water holding capacity in a grid cell. Total saturation excess runoff is partitioned into surface runoff and recharge to groundwater, with groundwater store being modeled as a linear reservoir to generate base flow. The open water body area within a cell is modeled separately, and total runoff
produced from the cell equals area-weighted runoff from land and open water bodies. The snow module does not distinguish open water area from land area within a grid cell.

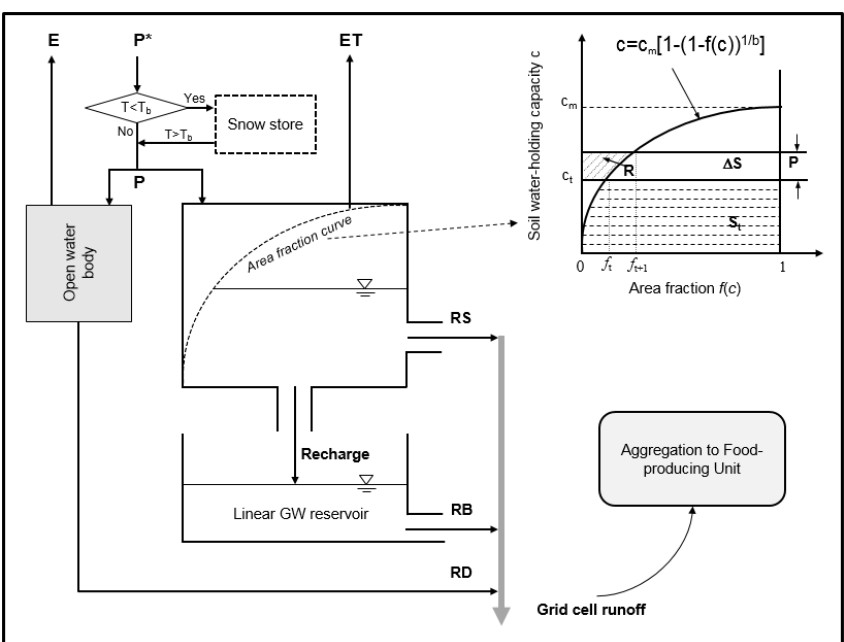

**Figure 1.** Schematic representation of the global hydrological model IGHM, illustrating vertical water balance of the land and open water fraction in a grid cell.
Note: $P^*$ = precipitation (mm/m); P = effective precipitation (mm/m); E = evaporation (mm/m); ET = evapotranspiration (mm/m); T = temperature (°C); $T_b$ = base temperature (°C), used as threshold to determine incoming precipitation as rain or snow; S = soil moisture content (mm); R = total runoff (mm/m); RS = surface runoff (mm/m); RB = base flow (mm/m); RD = direct runoff from open water body (mm/m).





The IGHM model as presented in this paper does not include a horizontal routing model. The remainder of this section describes the three modules in the IGHM model, which simulate PET, snow accumulation and melt, and runoff generation, respectively.

### 2.2.1 Potential evapotranspiration

Precipitation and evaporative demand are the two dominant climatic drivers of water balance. PET accounts for the availability of energy, but not moisture availability. In dry regions, actual evapotranspiration (AET) is limited by precipitation, whereas in wet regions it is limited by PET. Therefore, the choice of the PET method has a bigger influence for wet regions than dry regions in hydrological modeling. Weiß and Menzel (2008) compares four PET methods using gridded global climate data and concludes that the Priestley–Taylor equation proved to be mostly

suitable for a global application. The Priestley–Taylor equation is used in the WGHM model (Döll et al., 2003; Müller Schmied et al., 2014), which generates gridded runoff used to calibrate IGHM. Therefore, in IGHM we also use the Priestley–Taylor equation to calculate monthly PET, as follows.

$$\text{PET} = \alpha \frac{\Delta}{\Delta + y} (R_n - G) \qquad (1)$$

in which PET is in mm day$^{-1}$; $\alpha$ is assigned as 1.26 in a humid climate and 1.74 in an arid location with relative

humidity less than 60 percent in the month when peak evapotranspiration occurs, according to Shuttleworth (1993); $\Delta$ is the slope of the vapor pressure curve in kPa° C$^{-1}$; $\gamma$ is the psychrometric constant in kPa° C$^{-1}$; $R_n$ is net radiation at the land surface in mm day$^{-1}$; and $G$ is soil heat flux density in mm day$^{-1}$. Among these variables, $\Delta$ is a function of temperature, and net radiation includes downward shortwave radiation at the land surface and net longwave radiation. As explained in Section 2.1, downward shortwave radiation is from the WFDEI climate data. Unlike

downward shortwave radiation, monthly bias correction was not conducted for downward longwave radiation in WFDEI (Weedon et al. 2014). Net longwave radiation is calculated using air temperature, and downward shortwave radiation based on the method given in Shuttleworth (1993).

### 2.2.2 Snow accumulation and melting

A temperature-index approach is used in the IGHM model to relate snowmelt to air temperature, assuming snowmelt

only occurs when mean daily air temperature is above a threshold value, called base temperature (Anderson 2006; Gray and Prowse 1993). The depth of snowmelt water produced in time period $t$ is given by

$$M_t = \begin{cases} 0, & T \le T_b \\ \text{MF}_t \cdot (T_t - T_b), & T > T_b \end{cases} \qquad (2)$$

where $\text{MF}_t$ represents the melt factor, $T_t$ denotes the index temperature, and $T_b$ is the base temperature, which is set to 0 °C here. Seasonal variation of the melt factor is calculated following the Anderson Snow Model (Anderson

30 2006).

$$\text{MF}_t = \text{ND}_t \cdot [S_v \cdot (\text{MF}_{max} - \text{MF}_{min}) + \text{MF}_{min}] \qquad (3)$$





in which $ND_t$ is the number of days in time period $t$; $MF_{max}$ is the maximum melt factor assumed to occur on 21 June in the northern hemisphere and on 21 December in the southern hemisphere; $MF_{min}$ is the minimum melt factor assumed to occur on 21 December in the northern hemisphere and on 21 June in the southern hemisphere. We choose 3.5 mm/°C · day and 6 mm/°C · day for $MF_{max}$ and $MF_{min}$ respectively, following (Gray and Prowse 1993).

The seasonal variation parameter $S_v$ is given by

$$S_v = 0.5 \cdot \sin\left(2\pi \frac{J-J_0}{366}\right) + 0.5 \tag{4}$$

In Eq. (4), $J$ is Julien day number; $J_0$ is the Julien day number of spring equinox, assigned as 81.5 in the northern hemisphere, and 264.5 in the southern hemisphere, corresponding to 21 March and 21 September, respectively.

A mass balance method is used to track snow accumulation and snowmelt. For time period $t$, the snow water

equivalent at the end of the time period is given by

$$SN_t = \begin{cases} SN_{t-1} + P*_t, & T_t \leq T_b \\ SN_{t-1} - \text{Min}(M_t, SN_{t-1}), & T_t > T_b \end{cases} \tag{5}$$

### 2.2.3 Runoff generation

Soil water balance is simulated at each grid cell using a single layer water bucket concept. To represent sub-grid variability of soil water holding capacity $c$, we assume it spatially varies within each grid cell, following a parabolic

distribution function (Arnell 1999; Moore 1985; Wood et al. 1992; Zhao 1992; Zhao et al. 1980).

$$f(c) = 1 - \left(1 - \frac{c}{C_m}\right)^b \tag{6}$$

where $f(c)$ is the fraction of area in a grid cell with soil water holding capacity values below $c$; $C_m$ is the maximum soil water holding capacity value across all locations within the grid cell; and $b$ is the "shape parameter" that defines the degree of spatial variability of soil moisture holding capacity $c$.

The maximum amount of water that can be held in the grid cell is

$$S_m = \int_0^{C_m} [1 - f(c)] dc = \frac{C_m}{1+b} \tag{7}$$

In Fig. 1, $S_m$ equals the area between the parabolic curve and the x-axis, with area fraction values of the x-axis ranging from zero to one.

Assuming that at any time $t$, each point in the grid cell is either at $C_m$ or at a constant moisture state $c$, the mean areal

water storage $S$ associated with soil water holding capacity $c$ at time $t$ is

$$S_t = S_m \cdot \left[1 - \left(1 - \frac{c_t}{C_m}\right)^{1+b}\right] \tag{8}$$



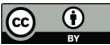

with precipitation $P_t$ and actual evapotranspiration $\mathrm{AET}_t$ in time period $t$, runoff is determined by the following equations (Wood et al. 1992; Zhao 1992):

If $c_t + P_t - \mathrm{AET} < C_m$,

$$R_t = P_t - \mathrm{AET}_t - \Delta S = P_t - \mathrm{AET}_t - S_m \cdot \left[ \left( 1 - \frac{c_t}{C_m} \right)^{1+b} - \left( 1 - \frac{c_t + P_t - \mathrm{AET}_t}{C_m} \right)^{1+b} \right] \tag{9}$$

Otherwise, if $c_t + P_t - \mathrm{AET} > C_m$,

$$R_t = P_t - \mathrm{AET}_t - (S_m - S_t) = P_t - \mathrm{AET}_t - S_m + S_m \cdot \left[ 1 - \left( 1 - \frac{c_t}{C_m} \right)^{1+b} \right] \tag{10}$$

The AET is determined jointly by the PET and the relative soil moisture state in a grid cell at time period $t$.

$$\mathrm{AET}_t = \mathrm{PET}_t \cdot \frac{S_t}{S_m} \tag{11}$$

The generated runoff in time period $t$ is divided into a surface runoff component RS and a deep percolation component
using a partitioning factor $\lambda$:

$$\mathrm{RS}_t = \lambda \cdot R_t \tag{12}$$

A linear reservoir is assumed to model base flow RB. The storage of the linear reservoir is linearly related to output, namely base flow, by a storage constant $\beta$ (Chow et al. 1988).

$$\mathrm{RB}_t = \beta \cdot G_t \tag{13}$$

In Eq. (13), $G_t$ is the groundwater storage value in time step $t$. The change of reservoir storage during time period $t$ is defined as

$$G_t - G_{t-1} = (1 - \lambda) \cdot R_t - \mathrm{RB}_t \tag{14}$$

The total runoff in time period $t$ is

$$R_t = \mathrm{RS}_t + \mathrm{RB}_t \tag{15}$$

The above equations (6)–(15) simulate runoff generation over the land portion of a grid cell. In many cells, total area of open water bodies in a cell accounts for a considerable fraction; for cells within the boundary of a large lake, the entire cell areas are covered by water. In these cases, water balance needs to be simulated separately for an open water body, as follows.

$$R_t^O = P_t - \mathrm{PET}_t \tag{16}$$





Equation (16) implies that an open water body evaporates at the rate of PET, and runoff is the difference between precipitation and PET. Runoff generated in an open water body area is negative when PET exceeds precipitation. For a cell that has both land area and open water body area, cell runoff is the area-weighted average of runoff values over land and open water body.

**2.3. Determination of model parameters**

Based on findings from previous studies, Wagener et al. (2001) concluded that up to six parameters can be calibrated from the time-series of external system variables (e.g., rainfall and runoff) using a single-objective calibration scheme. The IGHM model has four calibration parameters: the sub-grid variability shape parameter $b$, the total runoff partitioning parameter $\lambda$, the storage constant $\beta$, and the average soil water holding capacity $S_{\max}$.

We use gridded monthly runoff simulated by the WGHM model version 2.2b, driven by a homogenized WFD/WFDEI combination (homogenization details in Müller Schmied et al., 2016) for the time period 1980-1999 to calibrate the IGHM parameters on a cell-by-cell basis.

A genetic algorithm (GA) (Carroll, 2001) is used to automatically search model parameters that lead to the best overall model performance, defined by the objective function of the searching problem. Genetic algorithms have

been applied to the calibration of conceptual hydrological models in a number of studies. For instance, Wang (1991) applies a GA algorithm to the calibration of five water balance parameters and two routing parameters in a version of the Xinanjiang model. Nicklow et al. (2010) reviewed a number of hydrological model parameter identification studies that use a GA.

Here, the objective function in the GA calibration is to maximize the Kling–Gupta efficiency (KGE) (Gupta et al.

2009), as follows:

$$KGE = 1 - \sqrt{(1-r^2) + (1-\alpha)^2 + (1-\beta)^2} \qquad (17)$$

where $r$ is the linear correlation coefficient between simulated and observed runoff; $\alpha$ is the ratio between the standard deviation of the simulated and observed runoff; and $\beta$ is the ratio between the mean simulated and mean observed runoff. Thus, by maximizing the KGE we optimize model performance from a multi-objective perspective

by focusing on three separate criteria: correlation, variability error, and bias error (Gupta et al. 2009). KGE has been widely adopted as a model performance metric for global or basin scale hydrological modeling (Beck et al. 2016; Formetta et al. 2011).

In the GA algorithm, a population size of 50 and maximum generation of 51 are used. The number of years available for calibration tends to negatively affect calibration efficiencies, whereas it positively affects verification

efficiencies (Merz et al. 2009). In this paper, we choose the period 1980–1999 for model calibration and the period 2000–2009 for model validation.

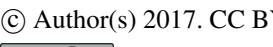



## 3 Results and Discussions

### 3.1 Model calibration and validation

Figure 2 shows gridded mean annual runoff over the global land surface, estimated from the IGHM simulation and from the WGHM 2.2b, respectively, for the period 1980–2009. The general consistency between the two mean

5   annual runoff maps can be clearly seen, suggesting that the IGHM is able to calculate renewable water resources at an annual time scale. In the IGHM, which has fixed open water body areas in each grid cell, negative runoff occurs in grid cells with open water bodies where evaporation exceeds precipitation. For WGHM, evaporation can be higher than precipitation (thus negative values for runoff) in the specific grid cell due to evaporation in surface water bodies which receive water from upstream.

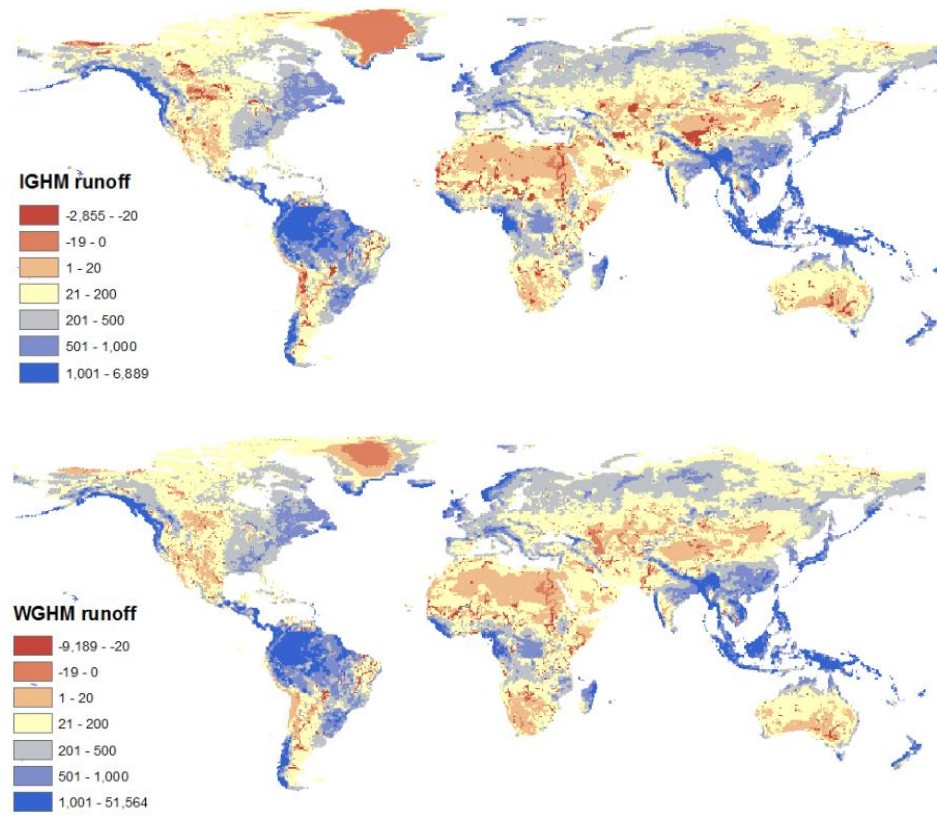

**Figure 2.** Comparison of simulated mean annual runoff during 1980–2009 between the IGHM and WGHM models, in mm/a.

To evaluate how well the calibrated IGHM model approximates "observations" globally, the KGE values for the

15   2000–2009 validation period and the 1980–1999 calibration period are shown in Fig. 3 respectively. The "observations" are gridded monthly natural runoff aggregated from daily natural runoff series simulated by WGHM





at the 0.5-degree resolution. A comparison of the KGE (validation) with the KGE (calibration) plot reveals that there is a strong linear correlation between the KGE of the calibration period and the KGE of the validation period; however, a high value of calibration KGE does not always result in a high validation KGE value. The cumulative distribution plots, also presented in Fig. 3, show that, in general, the IGHM simulations better reproduce WGHM

5    runoff in the calibration period than in the validation period.

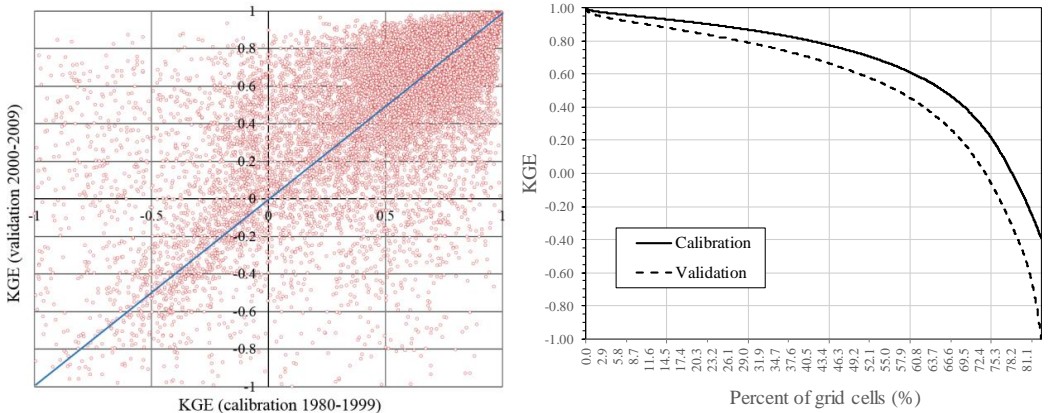

**Figure 3.** Cumulative distribution of KGE values in the calibration period (1980-1999) and validation period (2000-2009). The left figure excludes KGE values less than -1.0, and the right figure excludes KGE values less than -0.4 in the calibration period and KGE values less than -1.0 in the validation period.

The maps in Fig. 4 display global gridded KGE values for the 1980–1999 calibration period, and the 2000–2009 validation period, respectively. The KGE values are greater than 0.75 for the majority of grid cells, including South Asia, Southeast Asia, central and southern China, and southern Europe, where irrigation is intensive. The KGE values in Central Asia, the Middle East, and North Africa, where irrigation is also somewhat intensive, are mixed.

15    There are also a considerable number of grid cells where the KGE values are below zero. Notably, the KGE values are negative in a large continuous area in Canada and a large area in Russia, the latter of which largely overlaps with the Ob and the Yenisey river basins. Globally, we find that 66% and 58% of the 66,896 grid cells attain KGE values greater than 0.50 during the calibration and validation periods, respectively.

The Nash–Sutcliffe efficiency (NSE) (Nash and Sutcliffe 1970) are also calculated for the calibration and validation

20    period. Performance with NSE shows generally the same pattern as those of KGE shown on Fig. 4.

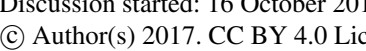



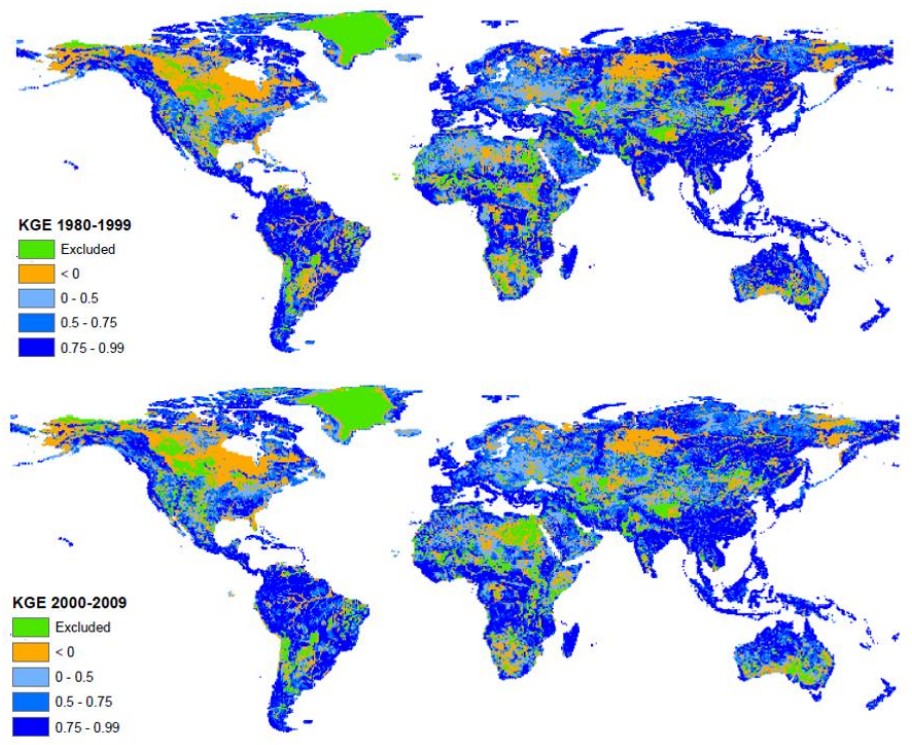

**Figure 4.** Kling-Gupta efficiency values in the calibration period (1980-1999) and the validation period (2000-2009). Mean annual runoff is estimated as negative in grid cells marked as "excluded" on both maps.

5    Figure 5 shows the components of KGE as given in Eq. (17), which include the linear correlation coefficient between simulated and "observed" runoff, mean annual runoff bias, and standard deviation bias, for the calibration period 1980–1999. The global map of correlation coefficients between simulated and "observed" runoff shows a similar spatial patterns as KGE. The correlation coefficient measures how well the model reproduces "observed" hydrographs, in particular the timing of peak runoff and low flow. From a water supply perspective (e.g., for

10    irrigation), a low correlation coefficient has a stronger impact for areas without adequate water storage infrastructure than those which have inadequate water storage infrastructure, due to the former's larger capacity to redistribute water temporarily to meet demand through, e.g., managing water supply or multi-purpose reservoirs. Correlations of monthly cell runoff are poor in particular in regions with many lakes and wetlands, like in Canada and the Ob basin in Siberia.




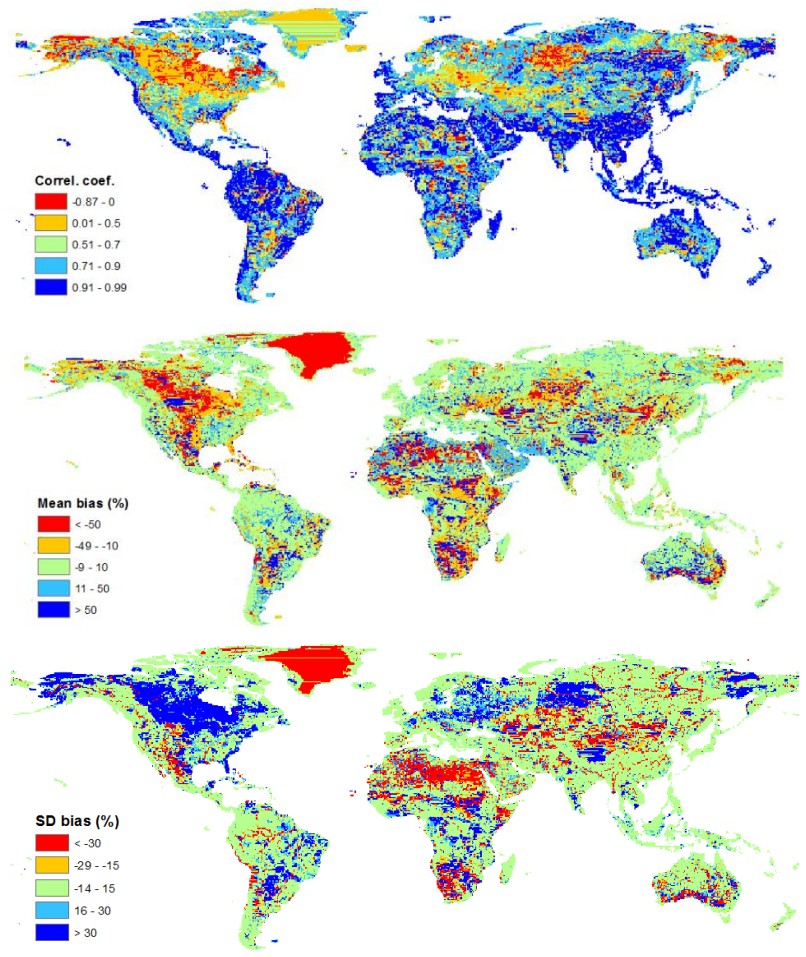

**Figure 5.** The model performance indicators in calibration period 1980–1999 including (a) linear correlation coefficient, (b) bias of mean annual runoff, and (c) bias of standard deviation.

The bias error (i.e., mean annual runoff bias) map shows that for the majority of grid cells, the bias error is within the range of 10% below or above the WGHM mean annual runoff. The bias error is larger for those grid cells that have a low KGE. The bias error is relatively high in some areas in the northern and southern Africa, and the central areas of North America. In the version of IGHM presented here, we do not apply a runoff correction factor.

10  The variability error (i.e., monthly runoff standard deviation bias) map also shows that for the majority of grid cells the variability error is within the range of ±15 % from the WGHM's standard deviation of monthly runoff series. Note that for much of the Asian monsoon areas, where variability is high both seasonally and interannually, the





variability error is kept within the ±15 % range. IGHM overestimates monthly runoff variability in lake and wetland areas like central Canada and the Ob basin and Yenisey basins in Russia and in Central and Eastern Europe. It is discernable that for river networks, notably the Amazon and rivers in Siberia on the variability error map, variability is underestimated. Since both the WGHM and the IGHM models use the Priestley–Taylor equation to estimate PET

and they use the same climate forcing data in PET and runoff calculation, the underestimation of standard deviations is likely caused by the different handling of surface water storages (lakes, wetlands) within each grid cell in the two models. The WGHM model considers storage (and area) variation in surface water bodies which lowers the monthly variability of runoff; while in the IGHM model, open water body areas in a grid cell is time-invariant.

### 3.2 Spatial patterns of calibrated model parameters

Since the IGHM model parameters are calibrated grid cell–by–grid cell, we expect to see a discernable spatial pattern of calibrated parameter values, which may be linked to biophysical attributes of grid cells. Fig. 6 shows the gridded values of the runoff generation shape parameter $b$, surface runoff fraction $\lambda$, inverse of groundwater residence time $\beta$, and effective soil water holding capacity $S_{\max}$.

The shape parameter $b$ defines the degree of variability in soil water holding capacity across a grid cell. In Fig. 1, the

distribution curve of soil water holding capacity is concave when $b$ is less than one, linear when $b$ equals one, and convex when $b$ is greater than one. Therefore, the higher the value of $b$, the higher the amount of runoff, assuming other parameters are fixed. The map of $b$ values in Fig. 6 shows that low values of $b$ (i.e., < 0.5) are widely distributed in dry areas, e.g., the Middle East and North Africa. The $b$ values in the rage of 1.1–1.5 are widely distributed in North America, Central America, South America, Europe, Africa south of the Sahara Desert, East

Asia, and Southeast Asia.

The runoff partitioning factor $\lambda$ can be influenced by land slope and soil infiltration capacity in a catchment. Its value is low in large areas in Canada, Europe, and central Africa. This implies that in these areas, river discharge does not respond rapidly to precipitation, and that base flow fed by groundwater makes a major contribution to river discharge. High values of $\lambda$ are found to distribute in dry areas, such as the southwestern United States, Mexico,

North Africa, Southern Africa, the Middle East, and Central Asia.

A higher groundwater residence time means that groundwater store is large, and base flow can last longer with a period of recharge. Therefore, the value of $\beta$ directly influences low flow simulation. For arid areas, such as in western North America, the Middle East and North Africa, Western Asia, and Australia, generally the value of $\beta$ is high. Groundwater residence time is short in these arid regions, resulting in very low or zero flow in dry months.






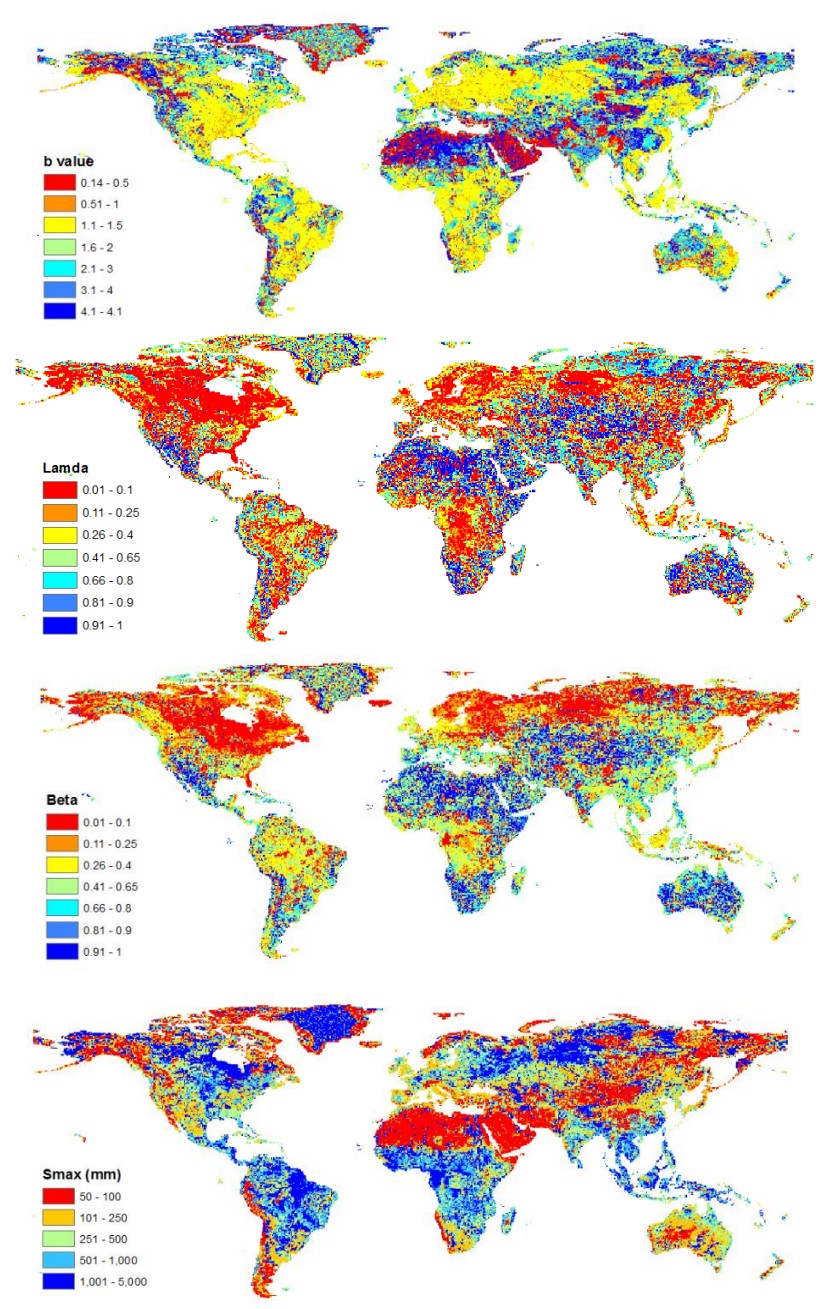

**Figure 6.** IGHM model parameters calibrated for the period 1980–1999.

The spatial pattern of soil water holding capacity $S_{max}$ appears to be related to climatic characteristics as given in the

5   world map of Köppen climate classification prepared by Peel (2007). For instance, the low value zones in North





Africa and the Arabian Peninsula largely overlap with the "arid-desert-hot (BWh)" Köppen climate zone. The BWh zones in the southwestern United States, western Mexico, and Australia also have a low value of soil water holding capacity. The tropical zones (tropical rainforest zone "Af", tropical monsoon zone "Am", and tropical savannah zone "Aw") in Africa, South Asia and Southeast Asia, Central America and South America generally have

moderately high to high $S_{max}$ values in Fig. 6. This overlap indicates that the way the parameter $S_{max}$ functions and the model structure to a certain extent reflect runoff generation processes in the real world.

The spatial patterns in Fig. 6 suggest a correlation between $S_{max}$ and the other three parameters. We conducted a spatial correlation analysis for each pair of the four parameters, and found that $S_{max}$ is negatively correlated with each of the other three parameters, as shown in Table 2. The spatial correlation of $S_{max}$ and b suggests that the

parameter that describes the degree of variability in soil water holding capacity may depend on the value of soil water holding capacity. This implies that, with the runoff generation scheme as applied to IGHM in this paper, the soil water holding capacity value at the grid cell level might be a useful factor for informing the estimation of the shape parameter. Similarly, the negative correlation between $S_{max}$ and $\lambda$ and between $S_{max}$ and $\beta$ suggests that in areas with low soil water holding capacity, the fraction of surface runoff is usually high, and groundwater residence

time (before becoming base flow) is usually low.

**Table 2.** Correlation between model parameters.

|   | $\lambda$ | $\beta$ | $S_{max}$ |
|---|---|---|---|
| **b** | 0.124 | 0.047 | –0.164 |
| **λ** | | 0.356 | –0.133 |
| **β** | | | –0.143 |

Note: Correlation analysis is conducted for each pair of the four calibration parameters on 66896 samples (i.e. grid cells). The t-test finds that these correlation coefficients are very highly significantly different from zero (P < 0.001).

## 4. Conclusions

A global hydrological model with a parsimonious structure is developed, calibrated, and validated in this paper. Using monthly climate forcing and runoff simulated by the WGHM model for the period 1980–1999, the IGHM model was calibrated grid cell–by–grid cell for this period, and was validated at the grid cell level for the period 2000–2009. Grid cell–specific KGE (and NSE) are calculated to measure model performance during both calibration and validation periods. We find that 66% and 58% of the 66,896 grid cells attain KGE values greater than 0.50

during the calibration and validation periods, respectively. For mean annual runoff, the model performs best in Asia, fairly well in Europe, Central America, the Caribbean, and South America, but less well in Africa, and the central longitudinal zone of North America. We noticed that the fixed open water body areas in IGHM may limit its capability to reproduce variability level (runoff standard deviation) for grid cells where the areas of open water body actively change due to rising and falling water levels, such as the grid cells along river networks.



We also find that the calibrated effective soil water holding capacity $S_{max}$ is influenced by climatic zones, with low values for the "arid-desert-hot" and high values in the "tropical" Köppen climate zones. The value of $S_{max}$ is negatively correlated with the other three model parameters, as identified by correlation analysis.

The primary application of the IGHM model is to compute monthly time series of water availability and to link to the existing water use simulation model in the IMPACT modeling system. It can also assess hydrological impacts of climate change, one of the key drivers in the long-term water and food projections in the IMPACT modeling system. The simulation of groundwater recharge in IGHM and its linkage to the water use simulation makes it possible to analyze groundwater use, depletion, and food production impacts in a dynamic manner. The parsimonious model structure and monthly time step are purposely chosen for the IGHM to meet the requirements of the monthly water use simulation and annual economic simulation in the IMPACT modeling system while at the meantime keeping the model structure relatively simple. As calibration parameters depend on applied climate data, recalibration may become necessary if better climate input data become available.

*Code and data availability.* The code of the IGHM model (Fortran 90) has been saved in CloudForge and is available upon request from the corresponding author. WFDEI (Weedon et al., 2014) is freely available at ftp://ftp.iiasa.ac.at/ (readme: ftp://ftp.iiasa.ac.at/README-WFDEI.pdf). WGHM data used for calibrating IGHM is available upon request (contact: P. Döll at p.doell@em.uni-frankfurt.de or H. Müller Schmied at hannes.mueller.schmied@em.uni-frankfurt.de).

*Competing interests.* The authors declare that they have no conflict of interest.

*Acknowledgement.* The research presented in this paper was supported by the CGIAR Research Program on Water, Land and Ecosystems.



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
