# Peer review of "Development and calibration of a global hydrological model for integrated assessment modeling"

_Geoscientific Model Development, 2017_

## Short Comment (SC1) · 25 Oct 2017

Dear authors,

In my role as Executive editor of GMD, I would like to bring to your attention our Editorial version 1.1:

http://www.geosci-model-dev.net/8/3487/2015/gmd-8-3487-2015.html

This highlights some requirements of papers published in GMD, which is also available on the GMD website in the 'Manuscript Types' section:

http://www.geoscientific-model-development.net/submission/manuscript_types.html

[Figure]

In particular, please note that for your paper, the following requirements have not been met in the Discussions paper:

- "The main paper must give the model name and version number (or other unique identifier) in the title."

- "If the model development relates to a single model then the model name and the version number must be included in the title of the paper. If the main intention of an article is to make a general (i.e. model independent) statement about the usefulness of a new development, but the usefulness is shown with the help of one specific model, the model name and version number must be stated in the title. The title could have a form such as, "Title outlining amazing generic advance: a case study with Model XXX (version Y)"."

- "All papers must include a section, at the end of the paper, entitled 'Code availability'. Here, either instructions for obtaining the code, or the reasons why the code is not available should be clearly stated. It is preferred for the code to be uploaded as a supplement or to be made available at a data repository with an associated DOI (digital object identifier) for the exact model version described in the paper. Alternatively, for established models, there may be an existing means of accessing the code through a particular system. In this case, there must exist a means of permanently accessing the precise model version described in the paper. In some cases, authors may prefer to put models on their own website, or to act as a point of contact for obtaining the code. Given the impermanence of websites and email addresses, this is not encouraged, and authors should consider improving the availability with a more permanent arrangement. After the paper is accepted the model archive should be updated to include a link to the GMD paper."

Thus the title should include the name (acronym) and version number of the hydrological model. Additionally, as you are using one specific integrated assessment model,

this should also be named (and versioned). E.g., "Development and calibration of a IMPACT Global Hydrological Model (IGHM vX.y) for integrated assessment modeling using IMPACT (v z.A)."

Additionally, please provide a reason why your model is not publicly available. It is scientific best practice that the code described in a paper is also made available to the public. Only legal issue might be a reason to sidestep this. Otherwise the publication has limited value. If the code /data are publicly available, please make the exact version, your article refers to, available via a permanent archive providing a DOI (e.g. Zenodo).

Yours,

Astrid Kerkweg

---

## Referee Comment (RC1) · Anonymous Referee #1 · 2 Apr 2018

This is a well-written paper describing the global hydrological component of the integrated assessment model IMPACT and its calibration. Generally, I have no major issues with this paper, except to say that certain integrated assessment models are actually moving to higher resolutions with all of their process descriptions and that they employ the original versions of global hydrological models (e.g. LPJml in IMAGE).

Therefore, it would be informative to compare the run times of WGHM and the simplified model IGHM (for instance per year integration) to see how much is gained by the use of the IGHM model. I think this is needed to underpin the rationale of this work.

Minor comments

[Figure]

Line 10, page 2: securtiy -> security

Line 20, page 2: uses -> use

Lines 4-5 and 10, page 6: first you speak of 7 challenges, and then of 2. This is confusing.

Line 20-25, page 6: The statement: "Second, hydrological models are traditionally developed based on measurements and understanding of "micro" scale processes. As such, observed data and hydrological processes are often not compatible or representative at larger scales relevant for macroscale processes (Singh and Woolhiser 2002)"

does not lead to the conclusion that:

"Therefore, sophisticated data-intensive watershed hydrological models may not be suitable for macroscale hydrological modeling, due to their large data requirements (Chen et al. 2007), the relatively highly detailed specifications of hydrological processes with a sophisticated model structure, and the large number of parameters that are tailored for a specific watershed at the cost of broader model applicability"

In fact, the conclusion from the first argument is that macro-scale hydrological models should be underpinned by correct upscaling procedures of parameters and processes to find a link with the scale of the project description (macro-scale) and that of the observations and process understanding (smaller scale). This upscaling may lead to a less complex model structure, but it does not have to be (if small-scale processes do not average out). Moreover, if the larger-scale model is simpler, it still does not have to be more parsimonious, because the data at the larger scale may be lacking to constraint the macro-scale parameters.

This (sometimes) false argument that simpler is necessarily more parsimonious keeps on popping up in the hydrological literature. A distributed model of a basin that is not calibrated but whose parameters are determined from auxiliary information that is available at that scale from DEM, soil map and remote sensing information, is more

parsimonious then a lumped conceptual model that has 7 free parameters that all have to be calibrated on a single hydrograph only.

Line 30-32, page 6: This argument is against physical logic. Usually, the more generally applicable a model or theory is, the more involved it is in terms of equations etc.

Line 7, page 7: priori -> a priori

Lines 1-2, page 8: natural runoff: are the reservoirs themselves taken out?

Line 7, page 8: "probabilistic distribution". It is better to speak of a spatial frequency distribution, because it represents spatial variation without actual reference to a specific location. It does not represent the outcome of some probabilistic process.

Line 8-10, page 10: modeling. "Weiß and Menzel (2008) compares four PET methods using gridded global climate data and concludes that the Priestley–Taylor equation proved to be mostly suitable for a global application"

This is not a strong argument. the main reason for using simpler PET relationships is the lack of data to parameterize e.g. Penman-Monteith (PM). However, we are 10 years down the road and much more datasets have become available since then. Also, PM has indeed problems in dry climates where the ventilation term may be too high because of lack of correct observations of RH in heterogeneous landscapes (feedback effects between land and atmosphere). However, Priestley-Taylor may underestimate evaporation and sublimation in colder areas during days with strong winds and little radiation.

Section 3.1: I understand that the main purpose of the model is to emulate WGHM. But it would also be good to have an idea about the "real" performance of the WGHM model used in this study, by showing some validation results of WGHM using GRDC data (or perhaps repeat some statistics from previous work and refer to this work).

Line 2, page 5: "strong correlation". Would be good to calculate its value and put it in the figure.

Figure 5: Also mask out the areas not considered such as in Figure 4. This would also allow you to increase the resolution of te legend.

Figure 6. The map for b in also has a magenta colour in it which is not in the legend.

Figure 6. Some of these parameters, such as Smax I expect to be part of WGHM as well. Thus, I would like to see some maps of these parameters compared to the patterns of similar parameters in WGHM to check for consistency of the calibration results.

Table 2. Apart from the correlation, it would be good to have a global sensitivity plot: global average KGE versus percentage change in each parameter.ÂăIs that possible? This would allow the reader to see which parameter has the largest effect on the calibration results.

---

## Referee Comment (RC2) · B. Jackson (Referee) · 15 Jun 2018

Given the length of time it is taking to secure a second review and given the positive nature of the first and my own initial impression of the manuscript, I have decided to move to a decision with one review along with another direct from myself as handling editor.

The paper is generally very well written with very few typos, I did not pick out any further to those already pointed out by the reviewer- except a question as to what is meant by the mm/a units for runoff – mm/year I assume? Please replace with something clearer.

[Figure]

There is a pressing need for computationally efficient models with this level of process detail and spatial representation to aid impact assessment and scenario analysis, so I concur this is a worthy contribution.

The extreme negative values (-3000 for IGHM and >9000 for WGHM) in Figure 2 are concerning – can you explain these? And discuss a little more the issues that might happen when you have a combination of open water and terrestrial land within a grid, particularly in arid conditions; lack of lateral transfers might be causing some artefacts here. It would be much preferable to split Figure 2 into one which gives the actual runoff values rather than runoff- evap, and additionally show the evap as an additional figure.

––––––––––––––––––––––––––––––––––

---

## Author Comment (AC1) · 16 Oct 2018

Dear authors, In my role as Executive editor of GMD, I would like to bring to your attention our Editorial version 1.1: http://www.geosci-model-dev.net/8/3487/2015/gmd-8-3487-2015.html This highlights some requirements of papers published in GMD, which is also available on the GMD website in the 'Manuscript Types' section: http://www.geoscientific-model-development.net/submission/manuscript_types.html

In particular, please note that for your paper, the following requirements have not been met in the Discussions paper: • "The main paper must give the model name and version number (or other unique identifier) in the title."

Response: We appreciate the Executive Editor for these comments. We have added the model name and version in the title. The revised title reads as "Development and calibration of the IMPACT Global Hydrological Model 2.0 (IGHM 2.0) for integrated assessment modeling using IMPACT 3.0"

• "If the model development relates to a single model then the model name and the version number must be included in the title of the paper. If the main intention of an article is to make a general (i.e. model independent) statement about the usefulness of a new development, but the usefulness is shown with the help of one specific model, the model name and version number must be stated in the title. The title could have a form such as, "Title outlining amazing generic advance: a case study with Model XXX (version Y)"."

Response: The model development relates to a single model and we now include the model name and version number in the title of the paper, as above explained.

• "All papers must include a section, at the end of the paper, entitled 'Code availability'. Here, either instructions for obtaining the code, or the reasons why the code is not available should be clearly stated. It is preferred for the code to be uploaded as a supplement or to be made available at a data repository with an associated DOI (digital object identifier) for the exact model version described in the paper. Alternatively, for established models, there may be an existing means of accessing the code through a particular system. In this case, there must exist a means of permanently accessing the precise model version described in the paper. In some cases, authors may prefer to put models on their own website, or to act as a point of contact for obtaining the code. Given the impermanence of websites and email addresses, this is not encouraged, and authors should consider improving the availability with a more permanent arrangement. After the paper is accepted the model archive should be updated to include a link to the GMD paper."

Response: Key elements of the IMPACT 3.0 model code were published as Robinson et al. 2015 and is available at https://www.ifpri.org/publication/international-model-policy-analysis-agricultural-commodities-and-trade-impact-model-0; IFPRI currently only makes the model code available to people trained on the integrated modeling system in international agricultural research centers.

Thus the title should include the name (acronym) and version number of the hydrological model. Additionally, as you are using one specific integrated assessment model, this should also be named (and versioned). E.g., "Development and calibration of a IMPACT Global Hydrological Model (IGHM vX.y) for integrated assessment modeling using IMPACT (v z.A)."

Response: The paper title has been changed to "Development and calibration of the IMPACT Global Hydrological Model 2.0 (IGHM 2.0) for integrated assessment modeling using IMPACT 3.0"

Additionally, please provide a reason why your model is not publicly available. It is scientific best practice that the code described in a paper is also made available to the public. Only legal issue might be a reason to sidestep this. Otherwise the publication has limited value. If the code /data are publicly available, please make the exact version, your article refers to, available via a permanent archive providing a DOI (e.g. Zenodo).

Response: Key elements of the IMPACT 3.0 model code were published as Robinson et al. 2015 and is available at https://www.ifpri.org/publication/international-model-policy-analysis-agricultural-commodities-and-trade-impact-model-0; IFPRI currently only makes the model code available to people trained on the integrated modeling system in international agricultural research centers.

---

## Author Comment (AC2) · 16 Oct 2018

Comment 1: This is a well-written paper describing the global hydrological component of the integrated assessment model IMPACT and its calibration. Generally, I have no major issues with this paper, except to say that certain integrated assessment models are actually moving to higher resolutions with all of their process descriptions and that they employ the original versions of global hydrological models (e.g. LPJml in IMAGE). Therefore, it would be informative to compare the run times of WGHM and the simplified model IGHM (for instance per year integration) to see how much is gained by the use of the IGHM model. I think this is needed to underpin the rationale of this work.

Response: First of all, we would like to thank the referee for these constructive and helpful comments that helped us to improve the quality of this paper. We have prepared responses to all comments as follows. Revisions made are highlighted in the text of the paper.

The run times of WGHM and IGHM differ substantially. For a 30-year simulation, with PET data pre-processed with the PET module of IGHM, the run time of IGHM runoff module is 1 minute 28 seconds on a Lenovo ThinkPad with Intel i7-5600U processor (base frequency 2.60 GHz), whereas the WGHM needs around 90 minutes for a 30-year simulation on a Delta server with i Xeon E5-1620 v4 processor (base frequency 3.50 GHz).

Changes in the manuscript: On page 6, Section 1.2, in the third paragraph, we added: "The reduction in computational time is significant when comparing runtimes of IGHM and WGHM. For a 30-year simulation, the run time of IGHM is 1 minute 28 seconds on a portable computer with Intel i7-5600U processor which has a base frequency of 2.60 GHz, whereas that of the WGHM is around 90 minutes on a Unix server with Intel xeon E5-1620 v4 (base frequency of 3.5 GHz)."

Minor comments

Comment 2: Line 10, page 2: securtiy -> security

Response: This has been corrected in the manuscript. We thank the reviewer for pointing this out.

Comment 3: Line 20, page 2: uses -> use

Response: This has been corrected. Thanks.

Comment 4: Lines 4-5 and 10, page 6: first you speak of 7 challenges, and then of 2. This is confusing.

Response: Thank you for pointing out this source of potential misunderstanding. The

seven challenges were raised in Döll et al (2016) in their paper about "challenges and prospects of global hydrological modelling". The seven challenges are listed in lines 6 to 9, page 6. Two of those seven challenges are of importance for the IGHM model. For example, the IGHM global hydrological model simulates natural runoff, without considering human water uses, so it is not affected by data scarcity for quantifying human water use.

Changes in the manuscript: Line 10 to 13 now reads as: "Out of the seven aforementioned challenges of global hydrological models, we identified two major challenges that hold true also for IGHM: data constraints and limited understanding of macroscale hydrological processes. These two challenges are…"

Comment 5: Line 20-25, page 6: The statement: "Second, hydrological models are traditionally developed based on measurements and understanding of "micro" scale processes. As such, observed data and hydrological processes are often not compatible or representative at larger scales relevant for macroscale processes (Singh and Woolhiser 2002)" does not lead to the conclusion that: "Therefore, sophisticated data-intensive watershed hydrological models may not be suitable for macroscale hydrological modeling, due to their large data requirements (Chen et al. 2007), the relatively highly detailed specifications of hydrological processes with a sophisticated model structure, and the large number of parameters that are tailored for a specific watershed at the cost of broader model applicability" In fact, the conclusion from the first argument is that macro-scale hydrological models should be underpinned by correct upscaling procedures of parameters and processes to find a link with the scale of the project description (macro-scale) and that of the observations and process understanding (smaller scale). This upscaling may lead to a less complex model structure, but it does not have to be (if small-scale processes do not average out). Moreover, if the larger-scale model is simpler, it still does not have to be more parsimonious, because the data at the larger scale may be lacking to constraint the macro-scale parameters. This (sometimes) false argument that simpler is necessarily more parsimonious keeps

on popping up in the hydrological literature. A distributed model of a basin that is not calibrated but whose parameters are determined from auxiliary information that is available at that scale from DEM, soil map and remote sensing information, is more parsimonious then a lumped conceptual model that has 7 free parameters that all have to be calibrated on a single hydrograph only.

Response: We thank the reviewer for these thoughts about model parsimony and do agree. We have revised the paragraph.

Changes in the manuscript: We revised the use of the term "parsimonious" and changed the paragraph to: "Out of the seven aforementioned challenges of global hydrological models, we identified two major challenges that hold true also for IGHM: data constraints and limited understanding of macroscale hydrological processes. These two challenges are interlinked. First, although new data sets and updated versions of existing data sets of climate, soil, and water bodies are being made available frequently, the representativeness and quality of these data sets are fundamentally limited by available in situ observations (Harris et al. 2014; Lehner et al. 2011). Second, hydrological models are traditionally developed based on measurements and understanding of "micro" scale processes, rather than macroscale processes (Singh and Woolhiser 2002)."

Comment 7: Line 30-32, page 6: This argument is against physical logic. Usually, the more generally applicable a model or theory is, the more involved it is in terms of equations etc.

Response: We agree with the reviewer and have deleted the sentence.

Comment 8: Line 7, page 7: priori -> a priori

Response: Corrected. Thanks.

Comment 9: Lines 1-2, page 8: natural runoff: are the reservoirs themselves taken out?

Response: Yes, in the WGHM model the flow regulation effects of dams and regulated

lakes have been switched off, thus handled as natural lakes. In the IGHM model, the lakes are simulated with water balance equation for open water body, as described by Eq. (16) in the paper.

Comment 10: Line 7, page 8: "probabilistic distribution". It is better to speak of a spatial frequency distribution, because it represents spatial variation without actual reference to a specific location. It does not represent the outcome of some probabilistic process.

Response: Thanks for pointing this out. We have replaced the word "probabilistic distribution" with "spatial frequency distribution."

Changes in the manuscript: On page 8, the original sentence has been revised to "... and determines total saturation excess runoff with a spatial frequency distribution of soil water holding capacity in a grid cell."

Comment 11: Line 8-10, page 10: modeling. "Weiß and Menzel (2008) compares four PET methods using gridded global climate data and concludes that the Priestley–Taylor equation proved to be mostly suitable for a global application" This is not a strong argument. the main reason for using simpler PET relationships is the lack of data to parameterize e.g. Penman-Monteith (PM). However, we are 10 years down the road and much more datasets have become available since then. Also, PM has indeed problems in dry climates where the ventilation term may be too high because of lack of correct observations of RH in heterogeneous landscapes (feedback effects between land and atmosphere). However, Priestley-Taylor may underestimate evaporation and sublimation in colder areas during days with strong winds and little radiation.

Response: We agree with the reviewer's note regarding the possible applicability of Penman-Monteith (PM) and limitations of both approaches. In addition to data requirements of PM (such as wind speed which is rarely available in reasonable quality at the model's resolution and global scale), we assume that the robustness of Priestley-Taylor with net radiation as main determine still has benefits compared to more complex approaches like PM (see also the discussion of Milly and Dunne 2017). However, the

main practical reason of using the Priestley–Taylor equation in IGHM is to enable consistency with the WGHM model which uses this approach.

Changes in the manuscript: This sentence/argument was deleted in the manuscript. The remaining sentences for justifying the use of the Priestley–Taylor equation read as (on page 9) "The Priestley–Taylor equation is used in the WGHM model (Döll et al., 2003; Müller Schmied et al., 2014), which generates gridded runoff used to calibrate IGHM. Therefore, in IGHM we also use the Priestley–Taylor equation to calculate monthly PET, as follows."

Comment 12: Section 3.1: I understand that the main purpose of the model is to emulate WGHM.

Response: Yes. The main purpose of IGHM is to emulate runoff values simulated by WGHM under the same climate forcing.

Comment 13: But it would also be good to have an idea about the "real" performance of the WGHM model used in this study, by showing some validation results of WGHM using GRDC data (or perhaps repeat some statistics from previous work and refer to this work).

Response: The performance evaluation of the WGHM model in a comparable (but slightly updated) version (see Müller Schmied et al. 2016a, 2016b, Müller Schmied 2017) has been done in the framework of the Inter-Sectoral Impact Model Intercomparison Project (ISIMIP), in which WGHM discharge values were compared against other global hydrological models and GRDC data globally. In summary, WGHM performs reasonably well in many regions and outperform other models in a majority of river basins, but have difficulties in e.g. cold climate zones (Zaherpour et al 2018). The work of Veldkamp et al. (2018) shows in particular that WaterGAP has no problems in capturing mean observed river discharge which is due to the calibration approach. The routing approach of WGHM is in some basins also comparable (or better) than a more physically based river routing approach (Masaki et al 2017). The model variant of

WGHM used in this study (without human impacts) does not allow meaningful comparisons to observed river discharge data as the large majority of river basins is affected by human impacts to some degree.

Changes in the manuscript: We added the following paragraph with regard to WGHM validation at the beginning of section 3.1: "Recent comparisons of a slightly different WGHM version (Müller Schmied et al, 2016a, b, Müller Schmied 2017) in the framework of the Inter-Sectoral Impact Model Intercomparison Project (ISIMIP 2a, https://isimip.org) showed a relatively good agreement with observed river discharge data for many basins; due to its calibration to mean annual river discharge, it outperforms other global hydrological models except in cold climate zones (Zaherpour et al 2018). The work of Veldkamp et al. (2018) shows in particular that WaterGAP has no problem in capturing mean observed river discharge, which is due to the calibration approach. Still, streamflow seasonality cannot be captured well in many river basins, in particular when streamflow is strongly affected by human water use and man-made reservoirs. The routing approach of WGHM is in some basins also comparable with (or better than) a more physically based river routing approach (Masaki et al 2017).WaterGAP monthly Nash-Sutcliffe efficiencies for streamflow at the 1319 gauging stations are larger than 0.7 for 372 stations, between 0.5 and 0.7 for 349 station and smaller than 0.5 for 598 stations (Döll et al, 2018)."

Comment 14: Line 2, page 5: "strong correlation". Would be good to calculate its value and put it in the figure.

Response: We did not find the word "strong correlation" in line 2, page 5. We checked the rest of the manuscript and found the following sentence on page 14, lines 2-3: "A comparison of the KGE (validation) with the KGE (calibration) plot reveals that there is a strong linear correlation between the KGE of the calibration period and the KGE of the validation period." This is the only sentence that mentions strong correlation throughout the paper.

Changes in the manuscript: Thus, we calculated the correlation coefficient and added it to the text that describes the figure (i.e. Figure 3).

Comment 15: Figure 5: Also mask out the areas not considered such as in Figure 4. This would also allow you to increase the resolution of te legend.

Response: Figure 5 was updated to mask out the areas not considered as in Figure 4.

Comment 16: Figure 6. The map for b in also has a magenta colour in it which is not in the legend.

Response: We double-checked Figure 6 and the way they (the four maps) were produced using ArcGIS. It appears that the maps do not include magenta color.

Comment 17: Figure 6. Some of these parameters, such as Smax I expect to be part of WGHM as well. Thus, I would like to see some maps of these parameters compared to the patterns of similar parameters in WGHM to check for consistency of the calibration results.

Response: We have added a map of Smax from the WGHM model.

Comment 18: Table 2. Apart from the correlation, it would be good to have a global sensitivity plot: global average KGE versus percentage change in each parameter.ÂËŸ aIs that possible? This would allow the reader to see which parameter has the largest effect on the calibration results.

Response: A sensitivity analysis figure was added to show how global average KGE values respond to perturbations in each calibrated parameter.

References:

Döll, P., F. Kaspar, and B. Lehner (2003), A global hydrological model for deriving water availability indicators: model tuning and validation, J. Hydrol., 270(1–2), 105–134, doi:10.1016/S0022-1694(02)00283-4.

Döll, P., T. Trautmann, D. Gerten, H. M. Schmied, S. Ostberg, F. Saaed, and C. F. Schleussner (2018), Risks for the global freshwater system at 1.5 °C and 2 °C global warming, Environ. Res. Lett., 13(4), doi:10.1088/1748-9326/aab792.

Masaki, Y., N. Hanasaki, H. Biemans, H. M. Schmied, Q. Tang, Y. Wada, S. N. Gosling, K. Takahashi, and Y. Hijioka (2017), Intercomparison of global river discharge simulations focusing on dam operation—multiple models analysis in two case-study river basins, Missouri–Mississippi and Green–Colorado, Environ. Res. Lett, 12(5), 55002, doi:10.1088/1748-9326/aa57a8.

Milly, P. C. D., and K. A. Dunne (2017), A Hydrologic Drying Bias in Water-Resource Impact Analyses of Anthropogenic Climate Change, JAWRA J. Am. Water Resour. Assoc., 53(4), 822–838, doi:10.1111/1752-1688.12538.

Müller Schmied, H., S. Eisner, D. Franz, M. Wattenbach, F. T. Portmann, M. Flörke, and P. Döll (2014), Sensitivity of simulated global-scale freshwater fluxes and storages to input data, hydrological model structure, human water use and calibration, Hydrol. Earth Syst. Sci., 18(9), 3511–3538, doi:10.5194/hess-18-3511-2014.

Müller Schmied, H. (2017): Evaluation, modification and application of a global hydrological model. Frankfurt Hydrology Paper 16, Institute of Physical Geography, Goethe University Frankfurt, Frankfurt am Main, Germany.

Müller Schmied, H., Adam, L., Eisner, S., Fink, G., Flörke, M., Kim, H., Oki, T., Portmann, F. T., Reinecke, R., Riedel, C., Song, Q., Zhang, J., and Döll, P. (2016a), Impact of climate forcing uncertainty and human water use on global and continental water balance components, Proc. IAHS, 374, 53-62, doi: 10.5194/piahs-374-53-2016.

Müller Schmied, H., Adam, L., Eisner, S., Fink, G., Flörke, M., Kim, H., Oki, T., Portmann, F. T., Reinecke, R., Riedel, C., Song, Q., Zhang, J., and Döll, P. (2016b), Variations of global and continental water balance components as impacted by climate forcing uncertainty and human water use, Hydrol. Earth Syst. Sci., 20(7), 2877–2898,

doi:10.5194/hess-20-2877-2016.

Veldkamp, T. et al. (2018), Human impact parameterizations in global hydrological models improves estimates of monthly discharges and hydrological extremes: a multi-model validation study, Environ. Res. Lett., 13(5), doi:10.1088/1748-9326/aab96f.

Zaherpour, J. et al. (2018), Worldwide evaluation of mean and extreme runoff from six global-scale hydrological models that account for human impacts, Environ. Res. Lett., 13(6), doi:10.1088/1748-9326/aac547.

---

## Author Comment (AC3) · 16 Oct 2018

Comments: Given the length of time it is taking to secure a second review and given the positive nature of the first and my own initial impression of the manuscript, I have decided to move to a decision with one review along with another direct from myself as handling editor.

Response: We appreciate the handling editor for conducting the review of this manuscript.

Comments: The paper is generally very well written with very few typos, I did not pick

out any further to those already pointed out by the reviewer- except a question as to what is meant by the mm/a units for runoff – mm/year I assume? Please replace with something clearer. There is a pressing need for computationally efficient models with this level of process detail and spatial representation to aid impact assessment and scenario analysis, so I concur this is a worthy contribution.

Response: Thanks for pointing this out. The runoff unit mm/a represents mm/year. We thank the handling editor for the positive comment on the need for computational efficient models.

Changes in manuscript: On page 13, in the title of Figure 2, we have changed the runoff unit to "mm/year."

Comments: The extreme negative values (-3000 for IGHM and >9000 for WGHM) in Figure 2 are concerning – can you explain these? And discuss a little more the issues that might happen when you have a combination of open water and terrestrial land within a grid, particularly in arid conditions; lack of lateral transfers might be causing some artefacts here. It would be much preferable to split Figure 2 into one which gives the actual runoff values rather than runoff- evap, and additionally show the evap as an additional figure.

Response: The extreme negative numbers (e.g. <-500 mm/year) in WGHM arise in few grid cells and are results of the calculation of large water bodies (located in dry regions where potential evapotranspiration PET is larger than precipitation P) in their outflow grid cell. Negative runoff values in WGHM indicate that inflowing water from upstream evaporates in lakes and wetlands in which the difference between PET and P exceeds local runoff production in the grid cell itself (e.g. Sudd Swamps). Similar patterns were found in the runoff values of IGHM. We decided to modify the legend text of the maps and now avoid showing those extreme high and low values to not confuse the readers. With regard to open water and terrestrial land coexisting in a grid, the IGHM and WGHM models calculate average evaporation of the grid using open

water evaporation value and land evapotranspiration value weighted by the fractions of open water and terrestrial land areas within the grid cell. The IGHM model does not include lateral transfers, while lateral water flows are simulated in the WGHM model at daily time intervals. In the IGHM model, runoff generated over open water equals rainfall subtracted by potential evaporation. The absence of lateral flow processes in the IGHM may lead to over- (e.g. when inflow exceeds outflow and lake area expands) or under-estimation (e.g. when inflow is less than outflow and lake area shrinks) of actual evaporation in open water areas. With regard to Figure 2, it does show runoff, not runoff minus evapotranspiration. We agree that it is a good idea to show evapotranspiration, too. Actual evapotranspiration of IGHM and WGHM are shown in Figure 2b, and runoff maps are shown in Figure 2a.

Changes in manuscript: We now use Figure 2a to show natural runoff and Figure 2b to show actual evaporation values in the paper. In addition, we added the following paragraph in Section 3.1: "Lateral water flows are simulated in the WGHM model at daily time intervals, whereas the IGHM model does not include lateral transfers. The absence of lateral flow processes in the IGHM may lead to over-estimation of actual evaporation if an open water area shrinks, or underestimation if it expands. In Figure 2, the large negative runoff values in WGHM arise in relatively few grid cells and are results of the calculation of large water bodies located in dry regions where potential evapotranspiration is larger than precipitation. There are generally more negative runoff grid cells in the IGHM runoff map due to fixed open water areas used in the IGHM database."